# Sequencing the Neurome: Towards Scalable Exact Parameter Reconstruction of Black-Box Neural Networks

## Abstract

Inferring the exact parameters of a neural network with only query access is an NP-Hard problem, with few practical existing algorithms. Solutions would have major implications for security, verification, interpretability, and understanding biological networks. The key challenges are the massive parameter space, and complex non-linear relationships between neurons. We resolve these challenges using two insights. First, we observe that almost all networks used in practice are produced by random initialization and first order optimization, an inductive bias that drastically reduces the practical parameter space. Second, we present a novel query generation algorithm that produces maximally informative samples, letting us untangle the non-linear relationships efficiently. We demonstrate reconstruction of a hidden network containing over 1.5 million parameters, and of one 7 layers deep, the largest and deepest reconstructions to date, with max parameter difference less than 0.0001, and illustrate robustness and scalability across a variety of architectures, datasets, and training procedures.

## 1 Introduction

The rapid rise of Deep Learning and Artificial Intelligence demands a deeper understanding of the inner workings of Artificial Neural Networks, with stakes higher than ever. Neural Networks are now used ubiquitously in every day life: from personalized movie recommendations to automated research assistance and portfolio management. The ability to precisely reconstruct a neural network, discerning the firing patterns of individual neurons solely through query access, is of central importance, with massive implications in safety, security, privacy, and interpretability. Such methods may even hold the key to eventually unlocking the inner workings biological neural networks.

Up until this point, due to the difficulty of the problem, practical results have been very limited. This is not fully surprising: In the general case, recovering the weights of a neural network is a hard problem(Berner et al.; Chen et al., 2022; Goel et al., 2017; Jagielski et al., 2020).

One approach is to relax the constraints, and instead of producing an exact weight reconstruction, these methods are satisfied with a generating high quality approximation of the model behaviour (Papernot et al., 2017; Jagielski et al., 2020), often called a substitute network (Chen et al., 2017). This is accomplished using similar techniques to knowledge distillation (Hinton et al., 2015), where the blackbox network takes on the role of teacher, and the substitute model the student. This type of approach is attractive due to its simplicity: the teacher provides information in the form of input-output pairs, and the substitute learns directly from this data. While this mode of investigation has proven fruitful in a wide range of settings, (Hu & Pang, 2021a;b) and architectures(Shen et al., 2022), it cannot provide an exact specification of a neural network, and is thus limited in its usefulness and the guarantees that it provides.

A second approach limits the problem in a different way: by focusing specifically on exact weight recovery of a specific type of Neural Network: Feed forward networks with ReLU activations (Nair & Hinton, 2010). The ReLU function has a distinct piece-wise nature, and identifying when this transition occurs in each neuron can allow for the parameters to be identified, up to an isomorphism. This idea has produced lots of theoretical work (Rolnick & Kording, 2020; Milli et al., 2019; Chen et al., 2021; Daniely & Granot, 2021; Bona-Pellissier et al., 2023; Petzka et al., 2020; Phuong &

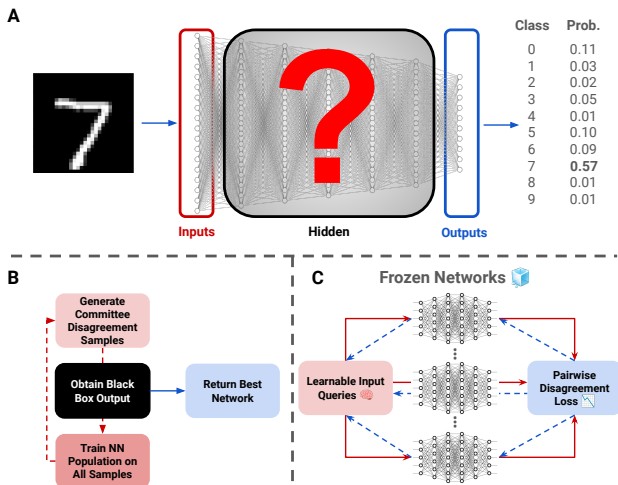

Figure 1: Problem Overview and Illustration of Reconstruction Algorithm and Query Generation Algorithm

Lampert, 2019), and recently has also led to some very recent strong empirical results (Carlini et al., 2020; Jagielski et al., 2020; Shamir et al., 2023). While these algorithms currently represent the state of the art in exact weight recovery, in practice these studies have only been applied to small networks, and reconstruction is a slow process that is fully limited to ReLU activations. While others have explored different analytic methods for inferring the weights, (Fiedler et al., 2023; Fornasier et al., 2019; 2022; Vlačić & Bölcskei, 2021), some of which can be applied to broader settings such as TanH networks, actual empirical results from all of these works have been very limited.

Of course, the most appealing approach would be to use the relatively simple and versatile methods of knowledge distillation, but to thereby precisely reconstruct the parameters of the network. However, directly training a student network to not just mimic the teacher, but converge on its exact parameters, is a very difficult proposition. Martinelli et. al. (Martinelli et al., 2023) proposes learning a larger substitute network, and then pruning it down to the proper size, although the resultant network usually will have more neurons than the blackbox and thus not be exactly identical. They went as far as to claim that, both from a theoretical and empirical perspective, directly learning the exact weights on a network of the exact same size as the black box is infeasible, and will inevitable get stuck in a high loss minima (Martinelli et al., 2023).

This work directly refutes this claim, and provides the first ever exact recovery of a neural network's full parameter set using the student teacher paradigm without extra neurons. Moreover, we show that our approach is actually well motivated by theory, and can solve larger, deeper, and more varied networks than previously seen in the literature. The best methods in the state of the art demonstrate exact reconstructions of up to 100k parameters on shallow ReLU and TanH networks, and up to 5 layers deep on a small ReLU network of roughly 1000 parameters (Carlini et al., 2020). We reconstruct networks with over 1.5 million parameters, and up to 7 layers deep, and demonstrate results on a variety of activation functions, network architectures, and training datasets.

We identify two main challenges in exact network reconstruction: navigating the massive parameter search space, and selecting informative queries that allow for sample efficient recovery.

Our approach to solving the first challenge is motivated by an important observation that has been almost entirely overlooked by previous work. While for the general case, reconstructing the parameters of a neural network is an NP-hard problem, we are primary not interested in solving for neural networks with arbitrary weight patterns, but in neural networks that are likely to exist in the real world. Because almost all networks are randomly initialized with a known distribution, and trained via backpropogation, the possible values that the parameters will practically take on is a minute subset of the full parameter space. To give an analogy from the field of image recognition: if previous algorithms have attempted to be valid for any possible configuration of pixels, we are proposing only considering pixel combinations likely to occur in real photographs.

To address the second challenge, we propose a novel sampling method called Committee Disagreement Sampling. From an information theory standpoint, the most useful sample is the one that evenly splits up the remaining parts of the hypothesis space that are consistent with the current samples (the version space )(Littlestone, 1988; Angluin, 1988). While generating maximally informative samples is NP-hard, it can be approximated using query by committee. This approach generates proxies for the most informative samples by selecting the samples that maximize disagreement among a population of hypotheses (Seung et al., 1992). Our sampling method generates new samples by generating random values and iteratively refining them using backpropagation to directly learn samples that maximize the disagreement of a population of potential solutions.

## 2 RESULTS

### 2.1 EXPERIMENT SETUP

Given a blackbox neural network, the goal is to reconstruct all of its internal parameters. We can query the network with any possible input and observe the corresponding output at the final layer. However, we have no access to any internal activations or weight values. A successful reconstruction extracts the parameters of the target network with a minimum number of input queries.

Like prior work(Carlini et al., 2020; Jagielski et al., 2020; Shamir et al., 2023), we assume exact knowledge of the target network architecture, including the number of neurons, their connectivity, and the activation functions. This is a reasonable assumption in practice because many companies and researchers publicly release the architectures of their trained models while keeping the exact trained weight values confidential(Brown et al., 2020; Touvron et al., 2023). Further, even when the architecture is not publicly released, side-channel attacks have been demonstrated that can infer this information(Hu et al.; Chabanne et al., 2021; Joud et al., 2023; Zhang et al., 2023).

Unlike other approaches (Truong et al., 2021), we will assume no direct or surrogate knowledge of the training dataset. However, we will make some assumptions about the training pipeline, namely that it uses standard procedures common in the training of modern neural networks. We will assume that all data was scaled to have a mean of $0$ and standard deviation of $0.5$, and that the network parameters were initialized with a mean of $0$ and standard deviation of

$$\sigma = \sqrt{\frac{2}{n_{in} + n_{out}}}.$$

where $n_{in}$ and $n_{out}$ are the number of incoming and outgoing neurons per layer (Glorot & Bengio, 2010). We further assume that the network was trained on the data using some form of gradient based first order optimization, although the exact optimizer, number of epochs, or learning rate, is not assumed, and can be anything.

### 2.2 ACCOUNTING FOR ISOMORPHISMS

Neural networks with different internal parameters can still exhibit the exact same input-output behavior. The input-output behavior of a network only defines its internal parameters up to an isomorphism, and depending on the architecture and type of activation functions used, different isomorphisms can be observed (Fiedler et al., 2021; Godfrey et al., 2022; Rolnick & Kording, 2020; Martinelli et al., 2023). Since two neural networks that are isomorphisms of each other are functionally identical, it is impossible to differentiate them using only query access, and thus when evaluating our solutions, we need to take these isomorphisms into consideration. There are three primary types of isomorphisms that are relevant for our network reconstructions:

**Permutations** Every neuron in a neural network computes an activation function over a linear combination of its input values. This linear combination implies that the order of the input values does not affect the computed result. Consequently, the input-output functionality of a neural network does not change when the internal order of the neurons changes. In other words, the order in which internal neurons are enumerated is arbitrary, and any two internal neurons can be swapped, as long as their connections to the preceding and next layer are preserved.

**Scaling** Networks with piece-wise linear activation functions, like ReLU and LeakyRelU, exhibit one more isomorphism: scaling. This is directly caused by the piece-wise linearity. Thus, for any positive scaling factor $\alpha$, the following holds:

$$f(\sum w_i \cdot (x_i \cdot \alpha) + b \cdot \alpha) = \alpha \cdot f(\sum w_i \cdot x_i + b)$$

This means that the output weights of a neuron can be scaled up as long as the input weights are scaled down with the same value.

**Polarity** Similarly, networks with an activation function symmetrical around zero, like TanH, exhibit another isomorphism of their own: polarity. For any input value, the activation function satisfies:

$$f(-x) = -f(x)$$

Therefore, the sign of the input weights of any neuron can be inverted when the sign of the output weights is inverted as well. Figure 4 illustrates the different isomorphisms.

When evaluating a solution, we search for the isomorphism of the solution that is most similar to the blackbox, and then compute parameter distance.

### CNNs

In convolutional neural networks, the isomorphism classes can be extended to convolutional kernels. The permutation, scaling, and polarity isomorphisms now operate over entire kernels and their associated output channels, rather than individual neurons. For permutations, the ordering of kernels within a convolutional layer is arbitrary, and any reordering of these kernels preserves the network's input-output behavior, provided that the corresponding input channels in the subsequent layer are permuted accordingly. For networks with piece-wise linear activation functions, all of the weights within any individual kernel can be scaled by a factor $\alpha$, as long as the corresponding input channel in the following layer's kernels is scaled by $\frac{1}{\alpha}$. Similarly, for activation functions symmetric around zero, the sign of all weights in a given kernel can be flipped, provided that the corresponding input channel in the next layer's kernels is also sign-flipped. This operation maintains functional equivalence due to the antisymmetry of the activation.

### RNNs

Recurrent neural networks introduce new isomorphism constraints due to the presence of recurrent connections through time. In particular, the hidden weight matrix is applied repeatedly over timesteps, and must be treated differently from standard feedforward weight matrices. Because the output of a recurrent layer at timestep *t* is saved as the hidden weight matrix and used in calculating the output at timestep *t+1*, the hidden weight matrix is effectively connected to itself, and any operation applied to its columns (interpreted as neurons) must also be applied to its rows (representing connections to other neurons). If neurons in the recurrent layer are permuted, the same permutation must be applied to both the rows and columns of the hidden weight matrix to maintain consistency. Similarly, if a neuron is scaled or polarity-flipped, both the corresponding column and row of the hidden matrix must be scaled or flipped by the same factor.

These additional constraints make RNN alignment substantially more complex. Because transformations must be applied jointly to both the rows and columns of the hidden matrix, the effect of any individual operation—scaling, polarity, or permutation—is entangled with all others. In other words, the transformation applied to any given weight depends not only on the operation applied to a single neuron, but on the combination of operations applied across all neurons simultaneously. This leads to a more global and coupled alignment problem than is encountered in feedforward or convolutional architectures.

## 2.3 RECONSTRUCTION EXPERIMENTS

To asses our algorithm, we began with a 3 layer 784x128x10 ReLU network with just over 100k parameters. This corresponds with the biggest network reported in prior state of the art methods (Carlini et al., 2020; Jagielski et al., 2020), although it is the smallest network that we will consider in our work.

When comparing to these prior SOTA methods, a few important things must be noted. First, while the work of Carlini et. al., the strongest existing method, is open source, we were unable to reproduce the results they reported. While the code worked for smaller networks, when we tried running it on our hardware to reconstruct the MNIST network with input dimension 784, or anything larger, it simply ran for several hours until crashing. Thus, we only provide comparison for the 784x128x10 MNIST network using the numbers reported in the literature, and do not provide direct comparisons on any of our other experiments. However, we encourage further experimentation, and thus we updated their code base to be compatible with the current version of JAX and included it with our code to facilitate comparison.

Second there are some slight architectural differences. We used LeakyReLU instead of ReLU to avoid dying neurons (Lu et al., 2019). We also used an output layer dimension of size 10, which is standard for MNIST classification, but they reported results on an output layer of only one dimension (784x128x1). Finally, our network used 32 bit precision, and they used 64. The comparison can be seen in Table 1.

| Reconstruction Method | # of Samples | max $\epsilon$ |
|---|---|---|
| Ours | 825k | 5.4e-05 |
| Carlini et. al. * | 2.9m | 1.4e-09 |
| Jagielski et. al. * | 1.2m | 2.8e-01 |
| Martinelli et. al. | N/A | 1.8e-04 |

Table 1: SOTA comparison on MNIST nets of size 784-128-10. Methods with * are limited to ReLU

Finally, while the table also includes results from Martinelli et. al. (Martinelli et al., 2023), it should be noted that this is not an exact reconstruction, since extra neurons were present. Their method also assumed knowledge of the training dataset.

We outperform all methods on sample efficiency (Martinelli assumed knowledge of the training dataset and thus did not use sampling), and compare well on max $\epsilon$ as well, especially when considering the one method to perform better used higher floating point precision.

| Blackbox epochs | # of samples | Max $\epsilon$ | Max $\epsilon$% | Mean $\epsilon$ per matrix |
|---|---|---|---|---|
| 5 | 550k | 4.3e-05 | 0.003% | 2.7e-06, 5.9e-06 |
| 25 | 550k | 5.4e-05 | 0.004% | 3.6e-06, 7.4e-06 |
| 50 | 550k | 6.3e-05 | 0.0045% | 4.4e-06, 7.5e-06 |
| 100 | 550k | 5.3e-05 | 0.004% | 5.0e-06, 7.1e-06 |
| 200 | 550k | 8.7e-05 | 0.006% | 6.2e-06, 6.9e-06 |
| 500 | 550k | 1.5e-04 | 0.01% | 8.3e-06, 6.5e-06 |
| 1000 | 1.1m | 5.4e-05 | 0.004% | 6.6e-06, 6.9e-06 |
| 5000 | 1.65m | 5.1e-05 | 0.004% | 5.6e-06, 7.2e-06 |

| Blackbox Optimizer | # of samples | Max $\epsilon$ | Max $\epsilon$% | Mean $\epsilon$ per matrix |
|---|---|---|---|---|
| ADAM | 550k | 5.4e-05 | 0.004% | 3.6e-06, 7.4e-06 |
| SGD | 550k | 5.4e-05 | 0.004% | 3.6e-06, 7.4e-06 |
| RMSPROP | 550k | 1.2e-04 | 0.0085% | 1.3e-05, 6.6e-06 |
| AdaDelta | 550k | 4.8-05 | 0.0035% | 2.1e-06, 7.7e-06 |
| Rprop | 550k | 4.2e-05 | 0.003% | 2.3e-06, 3.7e-06 |
| AdaGrad | 1.1m | 8.8e-05 | 0.006% | 9.3e-06, 6.8e-06 |
| Dataset | # of samples | Max $\epsilon$ | Max $\epsilon$% | Mean $\epsilon$ per matrix |
| MNIST | 550k | 5.4e-05 | 0.004% | 3.6e-06, 7.4e-06 |
| KMNIST | 550k | 3.6e-05 | 0.0025% | 3.1e-06, 6.4e-06 |
| Fashion MNIST | 550k | 8.7e-05 | 0.006% | 3.1e-06, 5.2e-06 |
| Cifar-10 | 2.75m | 5.1e-05 | 0.0035% | 2.2e-06, 8.2e-06 |
| Cifar-100 | 2.75m | 6.2e-05 | 0.0045% | 1.2e-06, 1.0e-05 |

Table 2: Reconstruction results as we vary blackbox training procedure for ReLU network of size 784x128x10. For Cifar-10 and Cifar-100, the network had to be modified to accommodate the larger image size, and thus consisted of an input layer of size 3072, and correspondingly 400k total parameters. We show mean error for each layer.

Since we are not assuming knowledge of the dataset, training duration, or optimizer, we evaluated our algorithm's robustness on a variety of scenarios. We varied the duration of the blackbox training procedure, experimenting on ranges of 5 to 5000 epochs. We also varied the optimizer that was used to train the blackbox network, sampling 6 of the most common ones. Finally, we varied the dataset that the blackbox was trained on. In all cases, the reconstruction method used to extract the blackbox parameters was exactly the same, with none of this information provided. We report max error between any two parameters. While max error is not a gameable metric, because of the presence of the scaling isomorphism described above, mean error can be manipulated by adjusting the scales of the two weight matrices such that the layer with fewer parameters has larger weight values, and the layer with more parameters has smaller weight values. To ensure the reader that we are not using scaling to manipulate the mean error, we report mean error for both weight matrices. We also report the max error as a percentage of the mean parameter magnitude, to give a sense of how small the errors are. The results can be seen in Table 2.

We further explored how our method scales with width and depth. A major result of the universal function approximation theorem (Leshno et al., 1993) is that networks of arbitrary width can express any continuous function. However, many studies have shown that the expressiveness of the network scales much faster with dept than with width(Lu et al., 2017; Safran & Shamir, 2017), and accordingly, we can expect deeper networks to be much more difficult to reconstruct. Our results represent both the deepest, and largest, exact reconstructions to date that we are aware of, as shown in Table 3.

| Architecture | # of Parameters | # of samples | Max $\epsilon$ | Max $\epsilon$% | Mean $\epsilon$ per matrix |
|---|---|---|---|---|---|
| 3072x128x100 | 406k | 2.75m | 6.2e-05 | 0.0045% | 1.2e-06, 1.0e-05 |
| 3072x256x100 | 812k | 5.5m | 7.5e-05 | 0.0055% | 1.7e-06, 1.1e-05 |
| 3072x512x100 | 1.6m | 5.5m | 9.2e-05 | 0.008% | 2.6e-06, 1.5e-05 |

| Architecture | # of Layers | # of samples | Max $\epsilon$ | Max $\epsilon$% | Mean $\epsilon$ per matrix |
|---|---|---|---|---|---|
| 784x128x10 | 3 | 3.3m | 4.5e-05 | 0.004% | 2.5e-06, 4.4e-06 |
| 784x128x64x10 | 4 | 3.3m | 1.0e-04 | 0.0085% | 1.9e-06, 4.9e-06,1.2e0-5 |
| 784x128x64x32x10 | 5 | 3.3m | 5.7e-05 | 0.004% | 1.3e-06, 2.3e-06, 5.6e-06, 1.2e-05 |
| 784x128x80x40x32x16x10 | 7 | 3.3m | 1.0e-04 | 0.006% | 1.2e-06,1.9e-06,3.3e-06, 8.0e-06,1.4e-05 ,1.3e-05 |

Table 3: Reconstruction across varied network widths and depths

To assess our algorithm's robustness beyond fully-connected networks, we applied it to a range of recurrent and convolutional architectures. Table 4 summarizes results on small to medium-scale CNNs with varying filter dimensions and depths, while Table 5 reports reconstruction errors for three RNN variants differing in hidden size. For small networks, we observe relatively low maximum deviations, demonstrating that our method can recover parameters across diverse model families, although reconstruction errors increase dramatically as model size and complexity increases. For these more complex models, increasing the number of samples seems to be a promising direction for exact reconstruction.

¿p3cm c c c c X

| CNN Architecture | # Params | # Samples | Max $\epsilon$ | Max $\epsilon$% | Mean $\epsilon$ per matrix |
|---|---|---|---|---|---|
| 1×3×3×3– 3×3×3×3– fnn10 | 1.6k | 550K | 5.75e-06 | 0.42% | 5.37e-07, 1.94e-07, 2.01e-07, 1.71e-07, 6.20e-07, 2.28e-07 |
| 1×10×3×3– 10×5×3×3– 5×3×3×3– fnn10 | 2.2k | 550K | 1.61e-05 | 0.01% | 1.55e-06, 7.67e-07, 3.12e-07, 4.75e-07, 2.68e-07, 1.45e-07, 2.18e-06, 4.41e-07 |
| 1×40×3×3– 40×20×3×3– 20×10×3×3– 10×3×3×3– fnn10 | 11.2k | 2.2M | 2.93e-01 | 430,200% | 3.45e-05, 1.97e-05, 1.55e-03, 4.19e-03, 1.60e-04, 2.34e-04, 1.27e-05, 9.98e-06, 7.83e-05, 1.51e-04 |

Table 4: Reconstruction across various CNN architectures.

l c c c c c X

| RNN Architecture | # Params | # Samples | Max $\epsilon$ | Max $\epsilon$% | Mean $\epsilon$ per matrix |
|---|---|---|---|---|---|
| rnn28×fnn10 | 1.9k | 1.1M | 1.02e-05 | 1.90% | 1.60e-06, 1.93e-06, 2.21e-06, 1.29e-06, 6.37e-07 |
| rnn64×fnn10 | 6.7k | 550K | 3.96e-04 | 103% | 2.72e-05, 5.71e-05, 2.12e-05, 9.21e-06, 4.89e-06 |
| rnn128×fnn10 | 21.5k | 4.4M | 7.25e-01 | 517,200% | 7.31e-02, 5.25e-02, 1.26e-01, 5.23e-02, 2.13e-04 |

Table 5: Reconstruction across various RNN architectures.

We demonstrate that our method generalizes beyond fully-connected networks to both recurrent and convolutional architectures of varying size and complexity. Although the reconstruction errors for large RNNs and deep CNNs are more modest, we are, to our knowledge, the first to successfully reverse-engineer any CNN or RNN purely through black-box queries. These results demonstrate the applicability of our algorithm across other network families and lay the groundwork for future exact reconstructions of more complex models.

## 2.4 CONVERGENCE ANALYSIS

To better understand how our algorithm converges, we performed an in depth analyis, using the most complex network we dealt with, the 7 layer 784x128x80x40x32x16x10 network trained on MNIST.

We looked at convergence per layer, as well as convergence per parameter. It is important to note that at iteration 25 we began relaxing the learning rate, which is why we see a discontinuity at that point.

There are several key takeaways. We can see that layers closer to the input converge first, and that, while the mean error gets low very rapidly, the max error in each layer takes far longer to converge. In the per parameter analysis, we plot every single network parameter, and can see the same phenomenon, where although the majority of errors are decreasing, a few pesky parameters stay with much higher error than the rest, as shown in figure 2.

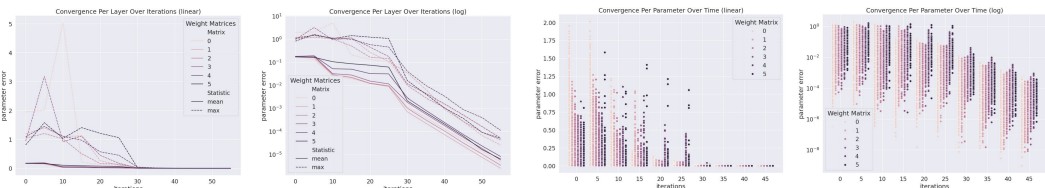

Figure 2: Illustration of Performance Convergence for 784x128x80x40x32x16x10 Network

We also measure how the reconstruction network converges to the functionality of the blackbox network. Input convergence was plotted as a series of heatmaps of size 28x28, the input space from MNIST. Each pixel represents the sum of all parameter errors that that pixel leads to, in every layer. Red represents larger error, black lower error. Ouput convergence was plotted per output neuron. Since MNIST has 10 classes (0-9), there are ten output neurons. We ran both the blackbox network and reconstruction through MNIST, and calculated the output difference average for each output Neuron, to represent in-distribution performance similarity. We should note that the reconstruction algorithm had no knowledge of MNIST.

The input behaviour shows that initially the error is highest on pixels towards the center. This makes sense, since in MNIST most semantic information is located in the center, and thus this is where the most complex weight behaviour is found. This error gradually decreases over iterations. The output behaviour shows convergence to near-identical behavior for all 10 classes. **Further, all of our reconstructions made the same classification as the blackbox network in 100% of cases**. Input and output convergence are shown in figure 3.

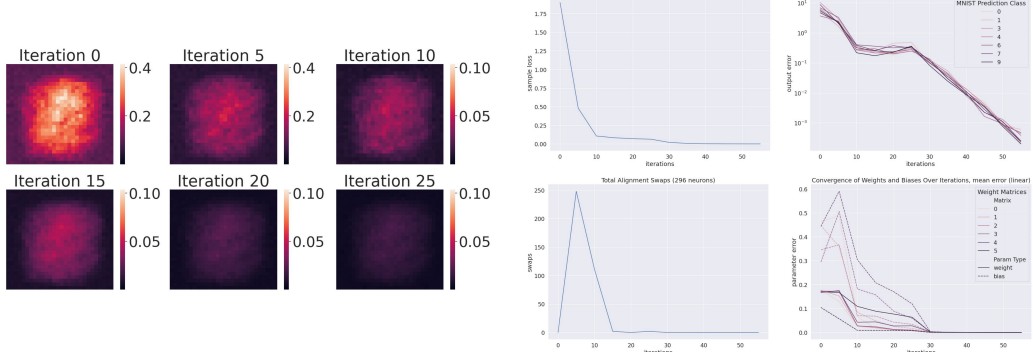

Figure 3: Illustration of Performance Convergence and Alignment Stability

We also compared convergence of weights and biases. This lead to an interesting result: the error of weight parameters decreases much more rapidly than biases, for every layer. We believe this is because bias behavior is more difficult to tease out by querying, since they are not multiplied by inputs. Finally we explored how stable our alignment algorithm remained as we approach convergence. We focused on the permutation isomorphism, and plotted how many times neuron alignments changed at each iteration. We can see in figure 3 that once mean weight error got below 0.05, the neuron alignment remained stable.

## 3 METHODS

### 3.1 RECONSTRUCTION ALGORITHM

Instead of trying to reconstruct any arbitrary network, as has been the focus of previous work, we focus on networks that have been produced via random initialization, and trained with gradient descent and backpropogation. There are several recent results that suggest this may be easier to solve than the general case. These ideas come from what has been called "The Modern Mathematics of Deep Learning", an area of analysis that emerged from trying to understand why neural networks seem to generalise so well and resist overfitting, even when heavily over-parameterized (Berner et al., 2021). This area of inquiry introduces several models that aim to describe how the weights of a neural network evolve during training.

While some alternatives have been proposed (Shi et al., 2024; Mukherjee & Huberman, 2022; Song et al., 2018), the Neural Tangent Kernel (NTK)(Jacot et al., 2018) is the most widely successful and adopted model, and it suggests that for over-parameterized networks, the weights barely move during training(Allen-Zhu et al., 2019; Du et al., 2019; Li & Liang, 2018). Chizat and Bak(Chizat et al., 2019) differentiate between the "lazy regime", where the weights barely move, and the non-lay regime (later dubbed the "rich regime" (Woodworth et al., 2020)) where the weights move a lot, and give conditions where lazy learning can occur even in small models. Li et. al. (Li & Banerjee, 2021) further demonstrated that even in the rich regime, the majority of parameters still exhibit lazy behaviour and barely move from their initial values.

While the NTK was first proposed for shallow feed forward networks, it has since been extended to deep networks(Lee et al., 2022), CNNs(Gu et al., 2020), RNNs (Alemohammad et al., 2020), GANs(Franceschi et al., 2022), Resnets (Huang et al., 2020), Auto-encoders (Nguyen et al., 2021), Transformers (Yang, 2020), and even decision trees(Kanoh & Sugiyama, 2022). Further, despite these studies being relegated to the realm of theory, often considering hypothetical network structures that cannot exist in practice, they do seem to model networks well in many real world cases (Seleznova & Kutyniok, 2022). In addition, while the usefulness of the NTK to describe the training dynamics breaks down as we train for longer, the Neural Collapse phenomena gives indication that even as training goes on for a long time, predictable features will emerge (Papyan et al., 2020).

It is also the case that networks trained using SGD, even with different random initializations, will tend to learn similar features(Gu et al., 2020), even across a variety of architectures (Mao et al., 2023) possibly a result of the so-called simplicity bias (Morwani et al., 2024; Wang et al., 2022), redundancy phenomenon (Doimo et al., 2022), symmetries(Głuch & Urbanke, 2021; Grigsby et al., 2023), and tendency of SGD to ignore certain minima(Barrett & Dherin, 2020)

The above results imply that, due to the inherent inductive biases of SGD, even after the training period, we still have strong priors of what the majority of the network weights will look like. In addition, a new model trained using SGD, is likely, at least under some circumstances, to find similar features to the original. This motivates that simply initializing a surrogate model of the same architecture as the blackbox, and trying to reconstruct the blackbox by sampling from it, and training the surrogate with a gradient based optimizer, is a strong candidate for exact weight recovery, assuming the black box itself was produced via gradient descent. Accordingly, our reconstruction algorithm is described in Appendix section B, Algorithm 1.

### 3.2 QUERY GENERATION ALGORITHM

Unlike in most modern ML settings, we know that the data we are training on was produced by a network of the same architecture as the substitute network, and thus a zero error hypothesis is guaranteed to be in our hypothesis space. Thus, it is logical to apply the result from the halving algorithm, that the most informative sample is the one that evenly splits the version space, and to approximate this using query by committee (Seung et al., 1992), as discussed above. This general setup is common in the active learning paradigm, where, just like in our case, we can arbitrarily query an oracle, but wish to minimize such queries (Settles, 2009), and is related to adversarial sampling in student-teacher distillation (Heo et al., 2019), except we do not have access to the internals of the teacher network.

Query by committee requires three ingredients:

1. The ability to construct a diverse committee

2. A disagreement criterion

3. A method of optimizing the queries over the disagreement criterion

While 1 is relatively straightforward via different random initializations, 2 and 3 are less obvious. Common methods suggested for 3 include hill climbing (Cohn et al., 1996), or simply just trying many samples and keeping the best ones. Inspired by the "hard sampling" method of Fang et. al.(Fang et al., 2019), we propose a novel sample generation algorithm that directly uses gradient descent to optimize the samples for maximal committee disagreement, along with a novel disagreement criterion that is generalizable to arbitrary length output vectors, and is continuous, so it can be optimized using gradient descent. Our query generation algorithm is described in Appendix section B, Algorithm 2.

This algorithms can be seen visually in figure 1.

Our disagreement criterion is defined as follows:

Let $I$ be a single input vector, of dimension $input\_dim$.

Let our population of networks $P$ that form the committee consistent of networks $N_1...N_p$.

We want to calculate pairwise disagreement among network outputs. We initially defined disagreement as L1-norm distance between outputs (Manhattan distance), but this led to a scaling issue, where our algorithm learned to cheat by realizing that simply having larger output magnitudes will produce a larger disagreement, even though nothing else has changed. This is especially a problem in ReLU networks, and led our algorithm to not learn anything useful. To rectify this, we first apply a normalization to each output vector by dividing each element by the vector's L1 norm. After normalization, we calculate the L1 distance between the vectors as the disagreement metric, solving the scaling issue. A more formal definition of disagreement loss is provided in Appendix section C

### 3.3 Aligning Networks

The neural network symmetries—permutations, scalings, and poarity flips—can be expressed as linear algebraic operations on weight matrices and convolutional kernels. In order to determine if two networks have been aligned, each layer can be normalized into a unique canonical form by scaling, negating, and permuting columns and rows (representing neurons and connections from neurons, respectively), of weight matrices. The exact methodology is further discussed in Appendix section D.

## 4 Limitations and Future Directions

Due to the stochastic nature of our method, there are times when it fails to work. For all experiments presented in this paper, the method was successful at least two thirds of the time, but there was not a 100 percent success rate for all networks.

In addition, further study is required to understand when and why this method fails. In particular, we note that narrow deep networks, while having a small fraction of the number of parameters of wide deep networks, were significantly harder to reconstruct and in a few cases failed.

Looking towards the future, we believe this study will be a powerful step towards exactly reconstructing full-sized real world networks. A large body of recent work demonstrates that for overparameterized networks, the weights barely move during training(Allen-Zhu et al., 2019). Chizat and Bak (Chizat et al., 2019) differentiate between the "lazy regime", where the weights barely move, and the rich regime where the weights move a lot, and give conditions where lazy learning can occur even in small models. Li et. al. (Li & Banerjee, 2021) further demonstrated that even in the rich regime, the majority of parameters still exhibit lazy behaviour and barely move from their initial values, and as training goes on for a long time, predictable features tend to emerge (Papyan et al., 2020). All of this evidence indicates that for larger networks, our prior assumption of random initialization and gradient based training provides an even stronger prior on the weight values, which is why we believe our approach is the best way to scale to larger networks.

## 5 REPRODUCABILITY STATEMENT

All of our work is fully reprodicable, including exact training scripts. Our code is included as a zip file in supplementary materials.

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

# A   IMPORTANCE AND APPLICATIONS

We believe this work is important for several reasons.

**Security** Knowledge of a network's structure is of central importance in adversarial machine learning (Huang et al., 2011). If we know the network parameters, we can attack it using gradient-based attacks (Papernot et al., 2016). While some attacks do not rely on such knowledge (Cisse et al., 2017), and there exists work that aims to make models robust to these types of attacks (Sitawarin & Wagner, 2019), this still represents perhaps the most significant attack vector for neural networks.

**Privacy** If we know the weights of the network, we can infer the training data (Haim et al., 2022; Fredrikson et al., 2015) which can be a severe privacy violation (Narayanan & Shmatikov, 2008), especially in medical domains (Loukides et al., 2010). Further, it may be undesirable for the weights of a network to be known. For example, large language models are very expensive to train, sometimes costing upwards of millions of dollars (Brown et al., 2020), and their owners may not want them being replicated.

**Interpretability** Another area where this analysis is useful is interpretability. As Deep Learning has become ubiquitous, the need for greater model interpretability has been stressed by many, for reasons ranging from ethics and legality to safety and security (Guidotti et al., 2018). The ability to reproduce a network's weights can give us insight into how it trains, what sorts of minima are common in networks trained via SGD, how subcomponents are related, and other aspects as well that can help reduce the black box affect.

**Safety** An additional important concern, related to the above, is the safety of a network for its users. An end user may commonly use a network provided by a third party for some important task, and relies solely on the guarantees of the third party that the network does what it purports to(Ahmad et al., 2021). The ability to reproduce the weights of the network can give users security and assure them that the network is safe to use, and opens up the possiblity of formal analysis of the parameters(Bona-Pellissier et al., 2023).

**Biological considerations** One of the greatest mysteries of the biological world is the human brain. Despite decades of research, much of its functionality is still not well understood. Reverse engineering biological neural networks is of foundational interest in neuroscience, and as has been noted by earlier work in this area (Rolnick & Kording, 2020), the ability to reverse artificial networks may give some insight into biological ones. Although there are many differences between artificial and biological neurons, neuroscientists have identified significant similarities, especially when zooming in to small regions (Kording et al., 2004), and many biological neurons appear to be well modeled by a ReLU artificial neuron (Chance et al., 2002). In fact, as early as 1981, similar experiments to the ones in this paper had already been conducted on biological neurons (Heggelund, 1981). While this is still a very far away thought, much like how sequencing the genome was a massive breakthrough brought about by steady incremental improvement (Heather & Chain, 2016), we believe work in this area will eventually contribute to our understanding of biological neurons.

## B  RECONSTRUCTION AND QUERY ALGORITHMS

---

**Algorithm 1** Reconstruction Algorithm

---

**Require:**
population size $p$, query number $q$,
outer iterations $o$, epochs $e$,
learning rate $\alpha$, schedule $S$
Empty dataset $D$
**Procedure:**
Randomly initialize a population of $p$ surrogate network with the same architecture as the target
network
**for** $o$ iterations **do**
    Produce $q$ samples and append them to dataset $D$
    **for** $e$ epochs **do**
        Train population on $D$ using learning rate $\alpha$
    **end for**
    **if** $o \in S$ **then**
        $\alpha \leftarrow \alpha/10$
    **end if**
**end for**
**Return** network in population with lowest loss

---

---

**Algorithm 2** Query Generation Algorithm

---

**Require:**
population $P$, query number $q$,
epochs $e$,
learning rate $\alpha$, schedule $S$
**Procedure:**
Randomly initialize a learnable tensor $I$ of shape $q$ x $input\_dim$
freeze the weights of $P$
**for** $e$ epochs **do**
    Forward-propogate $I$ through $P$, obtaining disagreement loss $DL_P(I)$
    Back-propogate loss and obtain gradient with respect to $I$
    Update $I$ using learning rate $\alpha$
    **if** $e \in S$ **then**
        $\alpha \leftarrow \alpha/10$
    **end if**
**end for**
**Return** $I$

---

## C  FORMAL DEFINITION OF DISAGREEMENT LOSS

We define a normalization function $f$, as $f(\mathbf{x}) = \frac{\mathbf{x}}{\|\mathbf{x}\|_1}$

The disagreement between two vectors, $u$ and $v$, is defined as

$d(\mathbf{u}, \mathbf{v}) = \sum_{i=1}^{n} |f(u_i) - f(v_i)|$

To get disagreement loss, we calculate the pairwise distance matrix between every network output
with every network output, for each network in the population.

$$D = \begin{bmatrix} d(N_1(I), N_1(I)) & d(N_1(I), N_2(I)) & \cdots & d(N_1(I), N_p(I)) \\ d(N_2(I), N_1(I)) & d(N_2(I), N_2(I)) & \cdots & d(N_2(I), N_p(I)) \\ \vdots & \vdots & \ddots & \vdots \\ d(N_p(I), N_1(I)) & d(N_p(I), N_2(I)) & \cdots & d(N_p(I), N_p(I)) \end{bmatrix}$$

Note, the diagonal here is 0, and the matrix is symmetrical around the diagonal, but this does not affect our calculation.

We then define the loss of input $I$ with respect to population $P$ as the negated mean of this matrix:

$$DL_P(I) = -\text{mean}(D) = -\frac{1}{p^2}\sum_{i=1}^{P}\sum_{j=1}^{p} D_{ij}$$

## D  ALIGNMENT ALGORITHM

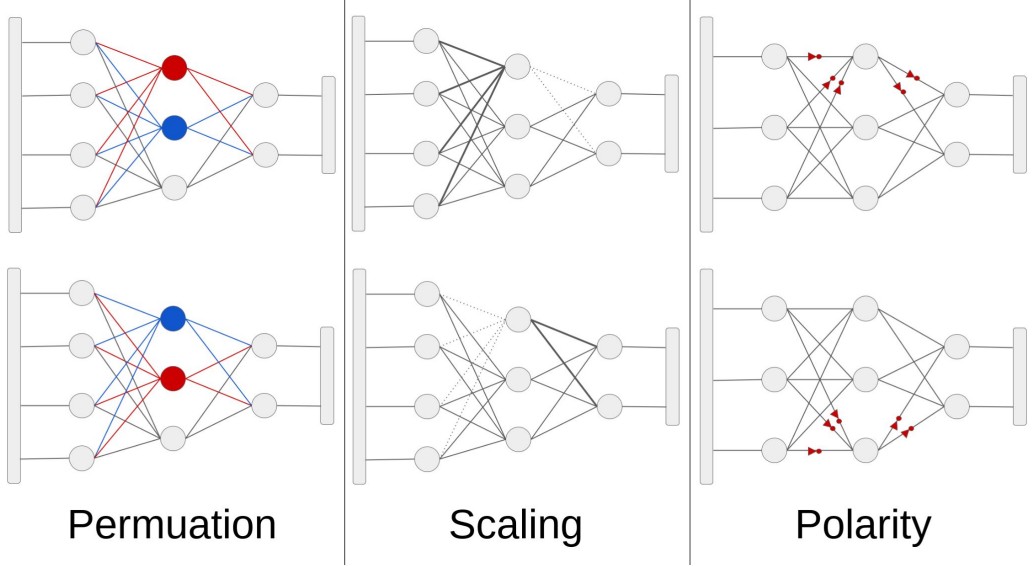

Figure 4: Illustration of Network Isomorphisms

We can mathematically describe the internal parameters of a neural network by enumerating the weight matrices of every layer in the network.

For the top left network in Figure 4, that would be:

$$\begin{bmatrix} w_{11} & w_{12} & w_{13} \\ w_{21} & w_{22} & w_{23} \\ w_{31} & w_{32} & w_{33} \\ w_{41} & w_{42} & w_{43} \end{bmatrix}, \begin{bmatrix} w_{11} & w_{12} \\ w_{21} & w_{22} \\ w_{31} & w_{32} \end{bmatrix} \tag{1}$$

Again, excluding the bias parameters for brevity. Similarly, for the bottom left network in Figure 4, we have:

$$\begin{bmatrix} w_{11} & w_{12} & w_{13} \\ w_{21} & w_{22} & w_{23} \\ w_{31} & w_{32} & w_{33} \\ w_{41} & w_{42} & w_{43} \end{bmatrix}, \begin{bmatrix} w_{11} & w_{12} \\ w_{21} & w_{22} \\ w_{31} & w_{32} \end{bmatrix} \tag{2}$$

In this representation, isomorphisms can be expressed as matrix operations. For example, swapping two neurons corresponds with swapping two column in a weight matrix and swapping the corresponding two rows in the subsequent weight matrix.

A similar observation can be made for the scaling isomorphism. The top middle network in Figure 4 can be represented with:

$$
\begin{bmatrix}
\alpha\mathbf{w_{11}} & w_{12} & w_{13} \\
\alpha\mathbf{w_{21}} & w_{22} & w_{23} \\
\alpha\mathbf{w_{31}} & w_{32} & w_{33} \\
\alpha\mathbf{w_{41}} & w_{42} & w_{43}
\end{bmatrix},
\begin{bmatrix}
\frac{\mathbf{w_{11}}}{\alpha} & \frac{\mathbf{w_{12}}}{\alpha} \\
w_{21} & w_{22} \\
w_{31} & w_{32}
\end{bmatrix}
\tag{3}
$$

And the bottom middle network in Figure 4 can be represented with:

$$
\begin{bmatrix}
\frac{\mathbf{w_{11}}}{\alpha} & w_{12} & w_{13} \\
\frac{\mathbf{w_{21}}}{\alpha} & w_{22} & w_{23} \\
\frac{\mathbf{w_{31}}}{\alpha} & w_{32} & w_{33} \\
\frac{\mathbf{w_{41}}}{\alpha} & w_{42} & w_{43}
\end{bmatrix},
\begin{bmatrix}
\alpha\mathbf{w_{11}} & \alpha\mathbf{w_{11}} \\
w_{21} & w_{22} \\
w_{31} & w_{32}
\end{bmatrix}
\tag{4}
$$

This example illustrates that the scaling isomorphism can be applied by:

1. scaling a weight matrix column with factor $\alpha$
2. scaling the corresponding row in the subsequent weight matrix with factor $\frac{1}{\alpha}$

Analogously, the polarity isomorphism can be applied by:

1. inverting the sign of a weight matrix column
2. inverting the sign of the corresponding row in the subsequent weight matrix

**CNNs**

We can similarly express the structural symmetries of Convolutional Neural Networks (CNNs) via transformations on their weight tensors. Each convolutional layer is represented as a 4-dimensional tensor of shape $(C_{\text{out}}, C_{\text{in}}, H, W)$, where each output channel corresponds to a 3D convolutional kernel applied across all input channels. These kernels can be considered as direct analogues of neurons in fully connected layers.

Let $[K_1, K_2, K_3]$ be a three-kernel convolutional layer, followed by a second convolutional layer containing a kernel $K_a$ of shape $3 \times 2 \times 2$. We can conceptually expand $K_a$ as follows:

$$
K_a = \left[ \begin{bmatrix} w_{111} & w_{121} \\ w_{211} & w_{221} \end{bmatrix}, \begin{bmatrix} w_{112} & w_{122} \\ w_{212} & w_{222} \end{bmatrix}, \begin{bmatrix} w_{113} & w_{123} \\ w_{213} & w_{223} \end{bmatrix} \right]
\tag{5}
$$

If the order of the kernels in the previous layer is permuted (e.g. $[K_1, K_2, K_3] \rightarrow [K_2, K_1, K_3]$), the corresponding input channels of each kernel in the following layer must also be permuted. For $K_a$, this results in:

$$
K_a = \left[ \begin{bmatrix} w_{112} & w_{122} \\ w_{212} & w_{222} \end{bmatrix}, \begin{bmatrix} w_{111} & w_{121} \\ w_{211} & w_{221} \end{bmatrix}, \begin{bmatrix} w_{113} & w_{123} \\ w_{213} & w_{223} \end{bmatrix} \right]
\tag{6}
$$

If all weights in $K_1$ are scaled down by a constant factor $\alpha$, the corresponding input channels of $K_a$ need to be scaled up by $\alpha$. For example, if the first layer has its first kernel scaled as $[\alpha\mathbf{K_1}, K_2, K_3]$, then $K_a$ must be scaled accordingly:

$$
K_a = \left[ \begin{bmatrix} \frac{\mathbf{w_{111}}}{\alpha} & \frac{\mathbf{w_{121}}}{\alpha} \\ \frac{\mathbf{w_{211}}}{\alpha} & \frac{\mathbf{w_{221}}}{\alpha} \end{bmatrix}, \begin{bmatrix} w_{112} & w_{122} \\ w_{212} & w_{222} \end{bmatrix}, \begin{bmatrix} w_{113} & w_{123} \\ w_{213} & w_{223} \end{bmatrix} \right]
\tag{7}
$$

Similarly to scaling, if the weights of $K_1$ are all negated, the corresponding input channel in $K_a$ must also be negated to maintain consistency.

**RNNs**

The isomorphisms in Recurrent Neural Networks are more nuanced due to the presence of weight sharing across time and the recurrence of hidden states. The hidden weight matrix $H \in \mathbb{R}^{n \times n}$ maps the output of the hidden layer at time $t$ to itself at time $t + 1$, effectively reusing the same set of

neurons as both source and target. Thus, any transformation applied on the columns of the hidden matrix must also be mirrored on its rows.

Let us consider an RNN layer with a hidden dimension of 3, followed by a feedforward output layer of dimension 2. We define:

$$W_{\text{in}} = [baseline=(m.base)] (m) [ matrix of math nodes, left delimiter = [, right delimiter = ], nodes = inner sep=2pt ] \text{w}_{11} w_{12} \text{…} \tag{8}$$

Swapping the first two neurons (i.e., applying a permutation isomorphism) requires permuting the columns of $W_{\text{in}}$ by swapping columns 1 and 2, permuting both the rows and columns of $H$ by swapping rows and columns 1 and 2, and permuting the rows of $W_{\text{out}}$ by swapping rows 1 and 2. The resulting matrices become:

$$W_{\text{in}} = [baseline=(m.base)] (m) [ matrix of math nodes, left delimiter = [, right delimiter = ], nodes = inner sep=2pt ] \text{w}_{12} w_{11} \text{…} \tag{9}$$

Now, consider the case of a polarity isomorphism, where we negate the first hidden neuron. This entails negating the first column of $W_{\text{in}}$, negating the first row and column of $H$, and negating the first row of $W_{\text{out}}$:

$$W_{\text{in}} = \begin{bmatrix} -\mathbf{w_{11}} & w_{12} & w_{13} \\ -\mathbf{w_{21}} & w_{22} & w_{33} \\ -\mathbf{w_{31}} & w_{32} & w_{33} \\ -\mathbf{w_{41}} & w_{42} & w_{43} \end{bmatrix}, \quad H = \begin{bmatrix} \mathbf{h_{11}} & -\mathbf{h_{12}} & -\mathbf{h_{13}} \\ -\mathbf{h_{21}} & \mathbf{h_{22}} & h_{23} \\ -\mathbf{h_{31}} & \mathbf{h_{32}} & h_{33} \end{bmatrix}, \quad W_{\text{out}} = \begin{bmatrix} -\mathbf{w_{11}} & -\mathbf{w_{12}} \\ w_{21} & w_{22} \\ w_{31} & w_{32} \end{bmatrix} \tag{10}$$

Note that $h_{11}$, the entry at the intersection of the first row and first column of $H$, is negated twice, once for each axis, and as a result remains unchanged.

## D.1 SIMILARITY OF NEURAL NETWORKS

When we calculate the similarity between two neural networks, we need to take these isomorphisms into account. We can do this by defining a canonical representation for each isomorphism group that is unique in every group. For every network, we define its canonical form as follows:

1. All weight matrix columns have unit norm, except for the last weight matrix.
2. All weight matrix columns have a positive sum, except for the last weight matrix.
3. All weight matrix columns are sorted according to their L1-norm, except for the last weight matrix.

(1) is only valid when the activation function is piece-wise linear and (2) is only valid when the activation function is symmetric around 0.

Now, we can calculate the similarity between two networks by converting both of them to their canonical form and calculating the sum of the L2-distances between their weight matrices.

Given a neural network we design the following procedure to convert it to its canonical form.

1. For i from 1 through N-1:
   (a) calculate the L2-norm of all columns in weight matrix i.
   (b) divide all columns by their L2-norm.
   (c) multiply the corresponding rows in weight matrix i+1 by the same L2-norm.
2. For i from 1 through N-1:
   (a) calculate the sign of the sum of all columns in weight matrix i.
   (b) multiply all columns by the sign of their sum.
   (c) multiply the corresponding rows in weight matrix i+1 by the same sign.
3. For i from 1 through N-1:
   (a) calculate the L1-norm of all columns in weight matrix i.
   (b) reorder the columns according to their L1-norm.

(c) reorder the corresponding rows in weight matrix i+1 accordingly.

Again, (1) is only performed for piece-wise activation functions and (2) is only performed for activation functions symmetric around 0.

While this procedure always obtains a distance metric of zero for isomorphic networks, it is not guaranteed to give a minimal distance value when two networks are not exactly isomorphic. Given two neural networks, we can apply a more exhaustive search to find the two representations that minimize the L2-distance between both networks. Because of the permutation isomorphism, this in an NP-hard problem. We devise a heuristic algorithm that runs in polynomial time by greedily matching weight matrix columns from both networks. The algorithm can be implemented by replacing the sorting step (3) in the above algorithm with the following matching step:

1. For i from 1 through N-1:

   (a) find a pair of a column from network 1 and a column from network 2 that has minimal L1-distance.
   (b) match this pair and remove both columns from the considered columns.
   (c) keep matching until all columns are part of a pair.
   (d) reorder the columns in network 2 according to the pairs that were discovered.
   (e) reorder the corresponding rows in weight matrix i+1 in the subsequent network.

We found that this procedure produces much more stable distance metrics when comparing two neural networks that are close but not identical, especially as the number of parameters grows. The disadvantage of this procedure is that it scales quadratically with the layer widths, compared to the sorting algorithm that scales linearly with the layer widths.

To align convolutional layers, we first transform it into its canonical form, following the analogous procedure for feedforward networks. This is justified by the observation that individual convolutional kernels play a structurally similar role to neurons in fully connected layers: each kernel defines a feature detector whose output is passed forward and linearly combined in subsequent layers. We find the canonical form of a convolutional network as follows.

For each convolutional layer, apply the following transformations:

1. **Scale Normalization** (if activation is piecewise linear): For each kernel in the layer, compute the Frobenius norm (i.e. L2 norm across all its weights). Normalize the kernel by dividing all weights by this norm. Then, multiply the corresponding input channel dimension in all of the subsequent layer's kernels by the same norm.

2. **Polarity Normalization** (if activation is symmetric around 0): For each kernel in the layer, compute the sign of the sum of its weights. Multiply each kernel by the sign of this sum. Then, multiply the corresponding input channel dimension in all of the subsequent layer's kernels by the same sign.

Once all convolutional layers have been transformed into this canonical form, we align the permutation isomorphisms by matching kernels between networks using the L1 distance, just as columns are matched between weight matrices in feedforward networks.

When aligning a convolutional layer that transitions into a fully connected layer, special care must be taken to properly propagate the scaling, polarity, and permutation transformations from the final convolutional layer to the first fully connected layer. This is nontrivial due to the flattening operation that bridges the two, reshaping the three-dimensional output feature map of the convolutional layer into a one-dimensional vector input to the feedforward layer. We use the insight that each kernel in the final convolutional layer contributes a contiguous block of flattened inputs (corresponding to rows in the fully connected weight matrix).

Let the final convolutional layer consist of $k$ kernels, each producing a feature channel of size $h \times w$. Upon flattening, the input to the feedforward layer becomes a vector of size $k \times h \times w$. Thus, this vector can be partitioned into $k$ contiguous index ranges, each of length $h \times w$, corresponding to the outputs from each kernel. The fully connected weight matrix is then of shape $(k \times h \times w) \times n$, where $n$ is the number of neurons in the fully connected layer. Each row in the weight matrix corresponds to a value in the flattened input vector, and thus each contiguous index range of rows in

the weight matrix can be mapped to a kernel in the previous convolutional layer. Thus, any scaling or polarity flipping of each kernel can have its reciprocal operation applied to each corresponding index range of rows in the feedforward weight matrix, and permutations can be applied by permuting the contiguous blocks of index ranges.

Aligning two recurrent networks presents additional challenges. For simplicity, we restrict our analysis to polarity and permutation isomorphisms, and assume that the nonlinearity in the RNN layer is the standard $\tanh$. In recurrent networks, special care must be taken with the hidden weight matrix. Any operation applied to the $i$-th column of this matrix must also be applied to the $i$-th row, because the same hidden state is both input and output at each time step. In other words, operations must be applied to row/column pairs indexed by $i$.

However, applying an operation (such as a sign flip) to the $i$-th row and column pair also indirectly affects all other row/column pairs. For example, consider the hidden matrix:

$$\begin{bmatrix} h_{11} & -h_{12} & -h_{13} \\ -h_{21} & h_{22} & h_{23} \\ -h_{31} & h_{32} & h_{33} \end{bmatrix}$$

Flipping the sign of the first row and column means the sign of the first element in the second and third row/column pairs also changes. Since each weight's sign depends on multiple overlapping operations, the effect of a single transformation is non-local. This interdependence makes it difficult to isolate the effect of a single operation, and therefore, the notion of a canonical form—as used for aligning feedforward and convolutional networks—does not directly extend to the hidden weight matrices in recurrent networks.

More specifically, for an $n \times n$ matrix, there are $2^{n-1}$ distinct polarity isomorphisms. (This arises because each of the $n$ row/column pairs can be either flipped or not flipped, giving $2^n$ possible combinations. However, flipping a subset of indices is equivalent to flipping the complement of that subset. As a result, each polarity configuration has a symmetric equivalent, reducing the total number of distinct isomorphisms to $2^{n-1}$.) Due to this exponential symmetry, identifying the "canonical form" of the matrix—i.e. finding the unique representative among all equivalent forms—is an NP-hard problem.

Thus, we perform a heuristic alignment based on the L2-norm of each column/row pair. Since the L2-norm of each pair is polarity and permutation invariant (flipping or reordering any of the weights still leads to the same L2-norm), we permute the hidden matrices in the order of sorted L2-norm for both network 1 and network 2. Then, we align the polarity of each column/row pair by choosing the polarity for each pair that minimizes the L1 loss between the two networks.

## E  ALTERNATIVE SAMPLING TECHNIQUES

To emphasize the importance of our query by committee generation strategy, we propose several logical sampling methods, and demonstrate how they fail to reconstruct the network. We divide sampling methods into two categories, non adaptive and adaptive. In the non adaptive setting, samples are generated without any knowledge of prior samples or of the current state of the reconstruction process. In adaptive sampling, samples are generated iteratively, with each new iteration making use of knowledge gained previously.

### E.1  NON ADAPTIVE SAMPLING

**Dataset Sampling** While our attack model does not assume knowledge of the original training data, it is logical to think that such knowledge may be useful for reconstructing the network, especially since Martinelli et. al. (Martinelli et al., 2023) demonstrated that for oversized substitute networks, this is sufficient. Thus, one method of non adaptive sampling is simply using the blackbox training dataset.

**Expanded Dataset Sampling** Similar to above, we make use of an extended real dataset larger than the original one used to train the black box, but still in a similar distribution. For our MNIST experiments, we do this by appending QMNIST, FashionMNIST, and KMNIST.

**Random Gaussian Sampling** Random sampling is the easiest form of sampling, and requires the least compute and domain knowledge. We considered random Gaussian sampling, with the mean 0 standard deviation 1.

**Random Uniform Sampling** We also considered random uniform sampling, with range [-1,1].

### E.2 ADAPTIVE SAMPLING

In addition to the above methods, we also considered adaptive sampling methods, where the samples we draw change based on what stage of the reconstruction process we are up to, and how well our hypothesis networks are fitting to the samples.

**Resampling Easy Regions** Borrowing easy and hard terminology from earlier work on sampling generators (Fang et al., 2019), we generate samples that are near the region where our network is approaching the target network functionality well

**Resampling Hard Regions** Here, we generate samples that are near the region where our network is predicting badly.

More specifically, we sample additional inputs as follows:

1. Calculate loss for all existing samples in dataset
2. Sort the losses from high to low
3. Find the $k$ samples with highest losses for hard sampling, and k lowest losses for easy sampling
4. Obtain the $k$ inputs corresponding with those $k$ samples
5. Recombine the components of the $k$ inputs into $n$ new inputs by random recombination of the feature values, with some small Gaussian noise added

Here we show the results of these sampling methods, and how none of them are able to be used to construct a network that matches the original, except for query by committee. For all sampling methods, we used 550k total samples, to make the comparison fair, except in Dataset and Expanded Dataset sampling, where we used the number of samples available.

| Sampling Method | # of Samples | max $\epsilon$ | Mean $\epsilon$ per layer |
|---|---|---|---|
| Original Full Dataset | 60k | 1.06 | 0.035, 0.024 |
| Expanded Dataset | 260k | 1.35 | 0.021, 0.015 |
| Random Gaussian | 550k | 3.4 | 0.004, 0.003 |
| Random Uniform | 550k | 3.2 | 0.005, 0.004 |
| Resampling Easy Regions | 550k | 3.3 | 0.007, 0.004 |
| Resampling Hard Regions | 550k | 3.2 | 0.009, 0.005 |
| **Committee (ours)** | **550k** | **4.4e-05** | **3.5e-06, 7.9e-06** |

Table 6: Failure of a variety of sampling methods, except query by committee, to solve a network of architecture 784x128x10.

## F RECOGNIZING CONVERGENCE

An important consideration in our algorithm is recognizing when we have converged, or if we are not converging. We have two reliable methods of doing this, and empirically, both have consistently worked, in the sense that every experiment that converged exhibited both properties, and every experiment that did not converge exhibited neither property. The two conditions are:

1. Population Agreement
2. Vanishing Loss

As we converge on a solution, several networks in our population will start to converge on the same weights. Once we have several members of the population reach identical weights, within a small epsilon, we can be sure our soluton has converged.

In addition, we can look at sample loss. As we converge, the L1 loss becomes vanishingly small, often in the range of $10^{-10}$, as show in in figure 3, and this always indicates convergence.

## G CONVERGENCE OF VARIOUS ARCHITECTURES

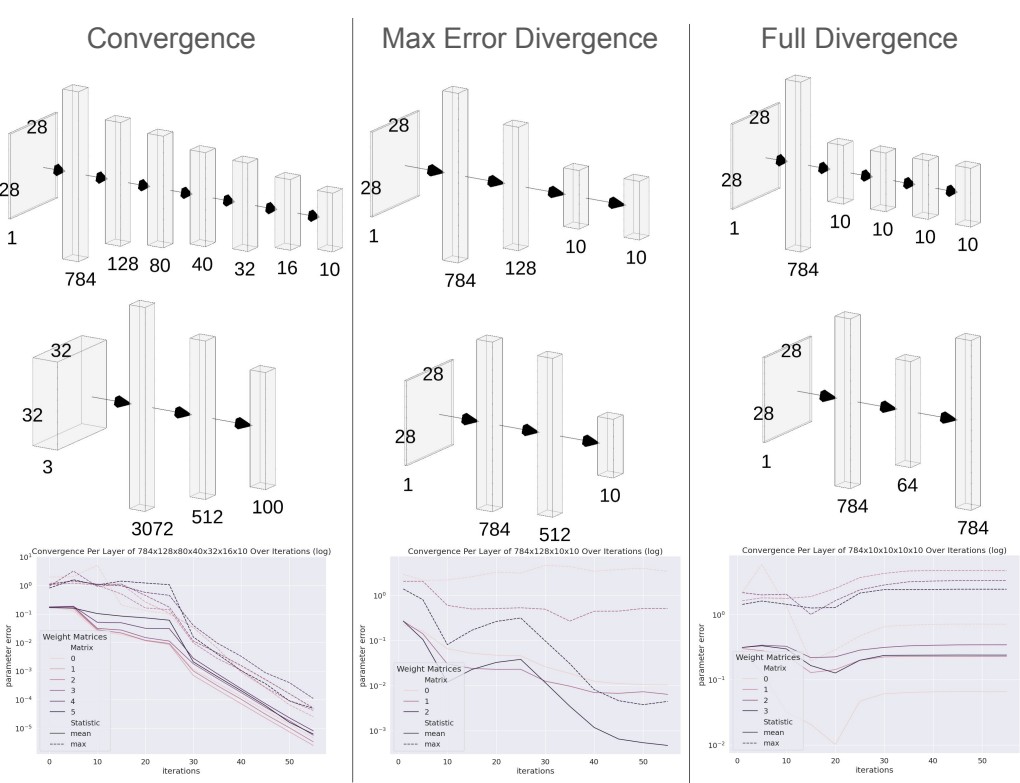

Figure 5: Illustration of Architectures, and their properties of Convergence and Divergence

We performed an analysis, as shown in figure 5 of which architectures did and did not converge. The conclusion was that pyramid networks, where the layer size is getting gradually smaller, were the easiest to reconstruct, and we were able to do so even for deep networks. However, if the network got narrow too quickly, or narrowed too slowly, reconstruction sometimes failed. Reconstruction was especially difficult for U-shaped networks. Also of note is that our algorithm experiences two kinds of divergence. The first, and most common, is that the mean error decreases gradually before tapering off, while the max error does not decrease. A more difficult form of divergence, common in deep but narrow networks shows even the mean error failing to decrease.

## H TANH ACTIVATION

We mentioned above that our algorithm works for other activations. Here we demonstrate this, and give results on networks using the TanH activation function.

## I ANALYSIS OF PRIORS

Our algorithm makes use of two priors:

| Architecture | # of Layers | # of samples | Max $\epsilon$ | Max $\epsilon\%$ | Mean $\epsilon$ per matrix |
|---|---|---|---|---|---|
| 784x128x10 | 3 | 3.3m | 6.0e-05 | 0.004% | 8.8e-06, 3.7e-06) |
| 784x128x64x10 | 4 | 3.3m | 7.4e-05 | 0.005% | 1.1e-05, 7.0e-06, 5.0e-06 |
| 784x128x64x32x10 | 5 | 3.3m | 1.0e-04 | 0.006% | 1.3e-05, 8.8e-06, 7.5e-06, 7.0e-06 |
| 784x128x64x32x16x10 | 6 | 3.3m | 1.0e-04 | 0.006% | 1.4e-05, 9.9e-06, 7.8e-06, 6.3e-06, 9.3e-06 |

Table 7: Reconstruction across varied network widths and depths

1. The assumption that weights do not move much during training

2. The assumption that we know the original weight distribution

Here, we explore what happens when we apply stronger versions of these priors. We devised two experiments.

**Untrained Network** We do not train the blackbox network at all. This represents a stronger version of the assumption that the weights did not move during training: here the weights did not move at all.

**Knowledge of Initial Weights** Instead of assuming we know the original weight distribution, we assume we know the original weights exactly. We experimented with two different ways of incorporating this knowledge. In one version, we initialized the entire committee population with the blackbox initial weights, and then added some small noise to give them variance. In the second method, we initialized a single network in the population with the original blackbox weights, and the rest of the population randomly.

Obviously, making both these assumptions at the same time renders the problem trivial, but independently they isolate our assumptions so that we can explore their significance.

### I.1 UNTRAINED NETWORK

It turns out that a fully untrained network is actually harder to solve than a trained one. This is because the outputs vary very little, and it is thus very difficult to tease out the weights via querying. However, we were able to validate our hypothesis somewhat, by demonstrating that a network trained for only a single epoch, where the weights barely moved, is indeed easier to reconstruct, as evident by the quicker convergence of the max errors in each layer. (We also note that, upon examining the code of Jagielski et. al. (Jagielski et al., 2020)), the network they reconstructed was trained on only a few dozen input samples)

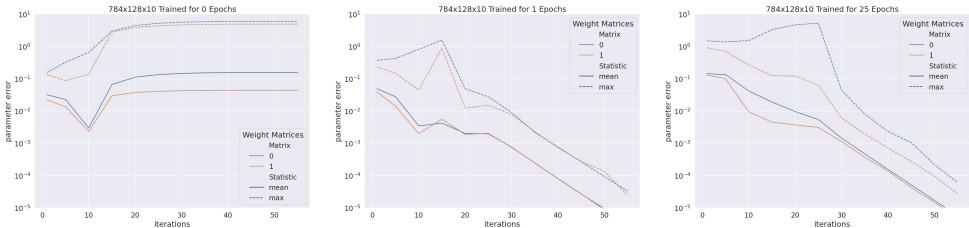

Figure 6: Reconstruction convergence as we train blackbox for different periods.

Table 2 above showed a similar idea: as we train for longer, eventually the number of samples required to reconstruct grows.

### I.2 KNOWLEDGE OF INITIAL WEIGHTS

When incorporating knowledge of initial blackbox weights, when we initialized the entire committee population with the blackbox initial weights, and then added some small noise to give them variance, we failed to solve at all, since the committee had too little diversity.

As a second attempt, we initialized only a single member of the population to the blackbox initial weights. This network did not converge faster than the randomly initialized networks.

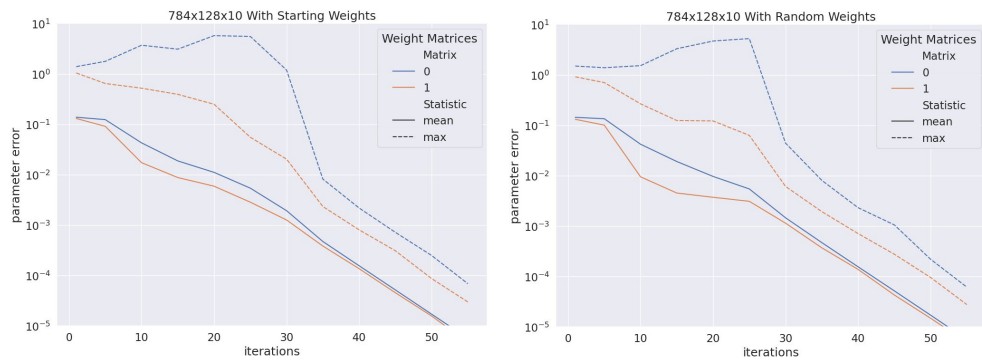

Figure 7: Knowing Initial Weights adds little convergence value

However, it was useful in a different sense. When running our algorithm, we developed a population of solutions as outlined above. When our algorithm converged, in general only part of the population would solve the problem, and the rest would get stuck in a local minima. The networks initialized with the original blackbox weights were much more likely to be in the part of the population that converged. This gives some insight into the importance of the initial population weights for reconstruction convergence.

## J    VISUAL OF COMMITTEE GENERATED SAMPLES

Here, we show heatmaps of our committee generated samples, at different iterations of the algorithm. Somewhat surprisingly, the samples still look like random noise, even after the networks have begun ton converge. This is somewhat logical, since our networks are likely to agree on simpler inputs.

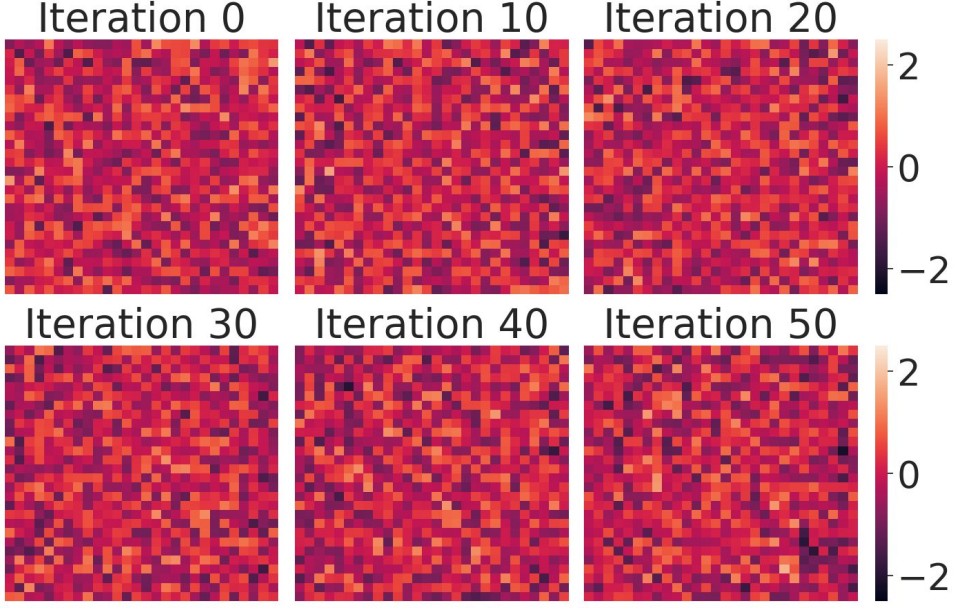

Figure 8: Illustration of Committee Generated Inputs

