# OpenReview forum: "Sequencing the Neurome: Towards Scalable Exact Parameter Reconstruction of Black-Box Neural Networks"
_ICLR.cc/2026/Conference — Submitted to ICLR 2026_

### Official Review · Reviewer_jd5e · 2025-10-21

**Soundness:** 3
**Presentation:** 2
**Contribution:** 3
**Rating:** 4
**Confidence:** 3

**Summary:**

This paper addresses the problem of exactly reconstructing neural network parameters solely from query access to a trained model. The authors build upon the student–teacher paradigm and claim to achieve the first-ever exact recovery of full network parameters without additional neurons, recovering models far larger than prior work (up to 1.5M parameters, 7 layers). The key innovations include: (1) Exploiting inductive biases from Glorot initialization and first-order optimization to reduce the search space. (2) Committee Disagreement Sampling, an active query strategy that uses backpropagation to learn inputs maximizing disagreement among candidate networks. The authors evaluate on various architectures (MLPs, CNNs, RNNs) and show substantially improved recovery rates compared to earlier reconstruction and model-extraction approaches. The paper also provides a detailed, albeit heuristic, treatment of network isomorphisms (symmetries) required for evaluation.

**Strengths:**

* **[S1] Ambitious problem.** The task of full parameter recovery of neural networks has implications for privacy, interpretability, and model security, making the research question highly relevant to the community.


* **[S2] Empirical scale and scope.** The experiments convincingly extend reconstruction to larger networks than prior work, indicating strong practical progress.


* **[S3] Well-motivated assumptions.** Building on Glorot initialization and gradient-based training reflects common practice and helps constrain the search space effectively.


* **[S4] Awareness of symmetry issues.** The discussion of some of the most prominent network isomorphisms (permutations, scaling) shows conceptual care and attention to the identifiability problem - the treatment is however heuristic due to the intractability of the problem.

**Weaknesses:**

* **[W1] Theoretical foundation.** The paper claims its approach is *"well motivated by theory"* (line 92) and repeatedly cites work on NTK and lazy vs. rich training. However, this is purely motivational. The paper provides no rigorous theoretical analysis of its own algorithm. There are no convergence guarantees, no sample complexity analysis, and no formal link between the cited theories and why this specific student-teacher disagreement algorithm should converge to the exact parameters. The *Convergence Analysis* in Sec 2.4 is purely descriptive and empirical.


* **[W2] Novelty.** The approach largely reuses ideas from query-by-committee (e.g., Seung et al., 1992). Using backpropagation to optimize queries for active learning is also not new. The novelty appears primarily empirical rather than conceptual. The paper needs to do a better job of situating its *Committee Disagreement Sampling* relative to these established ideas and pinpointing its specific, novel contribution.


* **[W3] Insufficient methodological exposition.** The paper is difficult to read due to its structure and lack of detail. The main text feels bloated with high-level motivation and citations of known results, while the actual algorithms (e.g. 1 and 2) that form the paper's core contribution are relegated to the appendix and are not discussed in detail. This makes it hard for the reader to grasp the method and follow the exposition.


* **[W4] Formatting issues.** Multiple typographical and citation errors, broke table formatting (e.g. Tables 4 and 5 are unreadable) hinder understanding.

* **[W5] Focus on Parameter-Level Recovery:** The paper focuses intensely on recovering exact parameters (modulo isomorphism). However, it doesn't clearly articulate why this is superior to achieving functional equivalence, which is the goal of most model extraction attacks. Is there a reason to care beyond evaluation?

**Questions:**

Please address the following questions and the above listed weaknesses in the rebuttal.

[Q1] Does the L1-based permutation alignment guarantee correct correspondence (I do not think it does), or might it actually overstate reconstruction accuracy? Taking a Bayesian stance here, the latter would happen with probability one for non-degenerate parameter distributions.


[Q2] In what precise way does the proposed method differ from classical query-by-committee and backprop-based query optimization? (Related to [W2])

[Q3] Is functional equivalence as described in Fig. 3 also given for the more complex tasks like CIFAR10? Please relate this also with [W5].

---

### Official Review · Reviewer_HGGv · 2025-10-29

**Soundness:** 2
**Presentation:** 1
**Contribution:** 2
**Rating:** 2
**Confidence:** 4

**Summary:**

This paper primarily explores how to recover the exact parameters of a neural network from query access. This problem is considered NP-hard but is crucial for areas such as security, interpretability, and understanding biological networks. The paper simplifies the problem by focusing on networks that are randomly initialized and trained through gradient optimization, effectively restricting the practical parameter space. It introduces a novel query generation algorithm, Committee Disagreement Sampling, aimed at generating maximally informative samples to efficiently untangle the complex nonlinear relationships in neural networks. The authors demonstrate the reconstruction of a hidden network with over 1.5 million parameters and a 7-layer deep network, with a maximum parameter difference of less than 0.0001. The paper shows that the method exhibits strong robustness in small-scale experiments across various architectures, datasets, and training procedures, outperforming existing methods in sample efficiency.

**Strengths:**

- The problem addressed in this paper is of significant value, as it tackles the challenging task of recovering the parameters of a network through black-box optimization.
- The paper introduces an  approach that leverages query-based methods and incorporates practical constraints from modern training processes to reconstruct the exact parameters of deep neural networks.
- This method has been validated on networks with over 1.5 million parameters and up to 7 layers deep. It has been tested across a variety of network architectures, datasets, and optimization procedures.

**Weaknesses:**

- The overall writing and formatting of the paper are quite poor, significantly impacting readability. There are numerous grammatical and typographical errors, making the paper appear as an unpolished draft (e.g. Table 4 and Table 5 ). The font size of Figure 2 is too small to be legible.
- The authors should discuss in more detail the significant differences between their approach and classic "distillation" techniques in LLMs.
- The evaluation metrics presented in the paper are difficult to understand. For example, in Table 1, the number of sampled samples varies, and the accuracy appears to be slightly worse compared to previous work. It is hard to grasp the superiority of the proposed method based on these metrics.
- While the experiments are conducted across various architectures, the datasets and models used are still relatively simple. I remain concerned about the scalability of the method, especially as it seems difficult to directly extend to current large-scale language models.
- The method assumes that the network architecture and various activation functions are known, but in practice, many hyperparameters are often unknown, which could limit its application in cases where the architecture is unknown or highly opaque.

**Questions:**

1. Could the authors elaborate on how their method differs from traditional knowledge distillation techniques, particularly in the context of large models with potentially billions of parameters? What unique advantages does their approach offer in such scenarios?
2. What is the precise definition of the error $\epsilon$ used in the paper? Given the issue of isomorphism, is it both sufficient and necessary to present the reconstruction error as the primary evaluation metric?
3. The evaluation metrics presented in the paper, particularly in Table 1, are somewhat difficult to interpret. The number of sampled queries varies, and the accuracy appears slightly worse compared to previous work. Can the authors clarify how the evaluation was conducted and explain why their method is considered superior, despite these discrepancies in sample size and accuracy? Would it be possible to provide a more detailed analysis or direct comparisons to prior work?
4. Could the authors offer insights into how the method might be adapted or scaled for larger models, particularly in the context of current large-scale language neural networks?
5. How does the method perform when the architecture of the target network is unknown or highly opaque? Are there modifications or adaptations that could make the method more applicable to black-box models where the architecture is not provided?

---

### Official Review · Reviewer_hac2 · 2025-10-30

**Soundness:** 2
**Presentation:** 1
**Contribution:** 2
**Rating:** 2
**Confidence:** 4

**Summary:**

This paper considers the problem of exact model reconstruction in a gray-box manner, assuming the knowledge of: (1) architecture, (2) data normalization, (3) the Initialization scheme of the trained network, and (4) full logit access. The proposed idea is to use a "committee" of $p$ student networks who have exactly the same architecture and initialization scheme as the reconstructed. At each time, the

**Strengths:**

The setting of this paper and their proposed method is easy to follow.

**Weaknesses:**

First things first, I admit that I am not an expert in this direction, so my review might be biased. However, I have many concerns with this paper as follows:

### Presentation weaknesses

1. Underwhelming presentation: I find the writing of this paper to be underwhelming, if not poor. For example:

  - Introduction: the authors should spend a separate paragraph, if not a section, elaborating their contribution. Besides, the storytelling in the introduction is also not well-written.
  - Results: right after the introduction, the authors go directly to demonstrating this result. This is a very weird flow, at least for me. The authors should at least: (1) have a formal problem setting, (2) state the assumption clearly and concisely, and compare with prior works about the setting. Then the author can go to their methodology, and then the results.
  - Methods: the method section is even mentioned AFTER the results section, making it very hard to keep track. I need to know what the authors are trying to do, what new insight/methodology the authors are proposing, before even looking at the results.

2. Many critical typos: apart from the presentation, I do not think that the author did a good job of screening the typos of this paper. For example, Tables 4 and 5 have a clear format error, though they are critical components of this paper.

### Technical weaknesses
1. Missing key components: the number of population $p$, which is a key component of their method, is mentioned NOWHERE in the paper. Without such information, it is very hard to evaluate the computational efficiency of this method. Besides, to make the sampling scheme of the proposed method to work reasonably, $p$ should be reasonably high. However, I assume that it would make the method very computationally inefficient, increasing at least $p$ times compared to using a single student.

2. The experiments are small-scale.

3. The results are not promising:
  - High error with even small models: even for small models, the error rate can be of 430,200% (for CNN) and 517,200% (for RNN) (see Tables 4, 5). This is very upsetting.
  - Highly unreliable: In the Limitation, the authors mentioned that the success rate is about two-thirds of the time. This is another critical weakness that cannot be treated as a "normal limitation". At least, the authors have to have some scheme to detect failure WITHOUT accessing the true model. If not, the proposed method will be very weak, and the reported results will also be unreliable.

These are the most critical issues of this paper that I can detect. Due to the reasons above, I unfortunately have to recommend a strong rejection of this draft. I recommend that the authors make multiple passes through this draft, make it at least look reasonable, and send it to their colleagues for cross-reviewing before even trying to submit this draft to any other conference.

**Questions:**

See Weaknesses.

---

### Official Review · Reviewer_UJ7W · 2025-11-01

**Soundness:** 2
**Presentation:** 2
**Contribution:** 3
**Rating:** 4
**Confidence:** 2

**Summary:**

The paper introduces a method for exactly reconstructing the full parameters of a neural network, given its architecture, initialization and optimization method, using only query access. Given these priors, the method assumes that the real weights lie in a smaller subspace of rand-init & sgd-trained networks, which can be recovered through gradient-based methods.
Reconstruction works by training a student network that attempts to match the teacher's parameters, via samples generated via committee disagreement sampling. Since isomporphisms manifest in the parameter space of neural networks, the authors normalize each layer into a canonical form, accounting for permutation, scaling and polarity symmetry. Evaluations show empirical reconstruction of an order of magnitude larger and deeper architectures (up to 1.6m parameters) compared to prior works.

**Strengths:**

* The paper sets out to achieve an ambitious goal, that of fully reconstructing model weights, with a solid theoretical motivation.
* Results presented in the paper showcase impressive reconstruction accuracy compared to the scale of prior art.
* The committee disagreement sampling method is an elegant method of generating the maximally informative samples for training the student model.

**Weaknesses:**

* The method is claimed to work only 2/3 of the time. How does an adversary know which is the working versions of the reconstruction?
    - Further exploring the failure modes would help mapping the impact and importance of the method.
    - Shedding more light in the dynamics of CNN and RNN networks would be a great plus.
    - Showcasing a proof-of-concept on a transformer architecture would also be important.
* The method requires significant computation for generating the training queries and results are mainly focused on toy-scale tasks.
* The results are mainly empirical and lack a formal proof of convergence of identifiability of a solution.
* The paper could significantly ameliorate its presentation:
    - Writing could be more standardized, instead of having the bulk of the method in the appendix.
    - No formal threat model present in the paper.
    - Figures are currently illegible and could use a better description of their setup.
    - The paper misses a dedicated section on how to defend against such attacks.

**Questions:**

* Does the method work on dynamic architectures (e.g., MoE models)?
* Is the reconstruction accuracy bound to the numerical representation of the parameters/activations (see quantization)?

**Details Of Ethics Concerns:**

The paper does raise a potential impactful technique of reconstructing model parameters, which can be an attack vector of IP. It fails to describe how it has ensured responsible disclosure and has also failed presenting defense mechanisms against such attacks.

---

### Official Review · Reviewer_HJTS · 2025-11-01

**Soundness:** 2
**Presentation:** 1
**Contribution:** 2
**Rating:** 2
**Confidence:** 2

**Summary:**

This submission studies exact parameter reconstruction of black box neural networks, assuming unconstrained query access. The proposed solution focuses in limiting the search space of possible reconstructions, taking into consideration inductive biases such as the model initialisation to a known distribution and 1st order optimisation through gradient descent, and introduces a generation algorithm for inference queries that aim to maximise the information extracted from each model invocation.

**Strengths:**

- The claim results indicate that the proposed approach may be able (under given assumptions) to accurately reconstruct the weights of larger models than prior work has explored thus far.
- Some interesting insights are presented and discussed in the experiments; that can motivate further development of future work.

**Weaknesses:**

- Although pushing the limits of weight reconstruction approaches in terms of scale, the proposed solution seems to remain computationally plausible only on toy-example scaled models. This limits the practical real-world applications of the proposed methodology.
- It is unclear whether the imposed assumptions on reconstructing randomly initialized models (with a known distribution) that undergone 1st order backprop training, are still applicable / can be relaxed on the most commonly setting of further finetuning models on domain specific data, following pre-training on larger scale datasets. This can severely affect the applicability (and thus impact) of the proposed approach in realistic scenarios.
- The manuscript does not examine at all the applicability of the proposed approach on Transformer architectures, nor discusses the challenges of this extension.
- The structure of the manuscript (section structure, content and titles) is quite irregular. e.g. the proposed solution is not clearly described anywhere other than the introduction section and some parts of the Appendix, and related work is mostly discussed in a section named "methods".

**Questions:**

Please consider replying on the comments raised in the weaknesses section above.

Minor comments:
- Tables 4 and 5 suffer from significant formatting issues.
- row 111: spacing issue in parenthesis.
- row 242: consider adding a reference, instead of referring to the method using the author's name directly.

---

### Meta-Review · Area_Chair_LjwB · 2025-12-16

**Summary:**

Reviewers unanimously agreed that the paper contains several points that require improvement prior to publication. These points of criticism ranged from issues surrounding presentation and writing to deeply technical ones. No responses, rebuttal, or pdf revisions were submitted in response to these concerns. As such, the AC recommends to reject the paper.

**Reviewer Concerns:**

There was no rebuttal or revision.

**Reviewer Scores:**

No rebuttals were submitted and thus initial reviewer scores would have been final scores as well.

---

### Decision · Program_Chairs · 2026-01-26

Reject